# Comparison of Marker Effects and Breeding Values at Two Levels at THI for Milk Yield and Quality Traits in Brazilian Holstein Cows

**DOI:** 10.3390/genes14010017

**Published:** 2022-12-21

**Authors:** Eula Regina Carrara, Brayan Dias Dauria, Izally Carvalho Gervásio, Robson Mateus Freitas Silveira, Gregori Alberto Rovadoski, Juliana Petrini, Mayara Salvian, Paulo Fernando Machado, Gerson Barreto Mourão

**Affiliations:** Department of Animal Science, “Luiz de Queiroz” College of Agriculture (ESALQ), University of São Paulo (USP), Piracicaba 13418-900, Brazil

**Keywords:** dairy cattle, DBV, fatty acids, genotype-environment interaction, heat stress, SNP

## Abstract

Genomic tools can help in the selection of animals genetically resistant to heat stress, especially the genome-wide association studies (GWAS). The objective of this study was to compare the variance explained by SNPs and direct genomic breeding values (DGVs) at two levels of a temperature and humidity index (THI). Records of milk yield (MY), somatic cell score (SCS), and percentages of casein (CAS), saturated fatty acids (SFA), and unsaturated fatty acids (UFA) in milk from 1157 Holstein cows were used. Traditional breeding values (EBV) were determined in a previous study and used as pseudo-phenotypes. Two levels of THI (heat comfort zone and heat stress zone) were used as environments and were treated as “traits” in a bi-trait model. The GWAS was performed using the genomic best linear unbiased prediction (GBLUP) method. Considering the top 50 SNPs, a total of 36 SNPs were not common between environments, eight of which were located in gene regions related to the evaluated traits. Even for those SNPs that had differences in their explained variances between the two environments, the differences were very small. The animals showed virtually no rank order, with rank correlation values of 0.90, 0.88, 1.00, 0.88, and 0.97 for MY, CAS, SCS, SFA, and UFA, respectively. The small difference between the environments studied can be attributed to the small difference in the pseudo-phenotypes used between the environments, on-farm acclimation, the polygenic nature of the traits, and the THI values studied near the threshold between comfort and heat stress. It is recommended that future studies be conducted with a larger number of animals and at more extreme THI levels.

## 1. Introduction

Hot and humid environments can cause heat stress situations in cattle, inducing behavioral and metabolic changes that negatively impact the animal’s health and milk production. For example, heat stress can lead to a reduction in feed intake and an increase in the respiratory rate and peripheral blood circulation of cows, in addition to causing changes in rumination activity [1,2]. Thus, it becomes important to improve the understanding of the interaction between the genotype and the environment for the consequent selection of animals more adapted to such challenging conditions. Several studies report variations in the estimates of variance components with consequent variation in the estimates of genetic parameters when milk production and quality traits are evaluated as a function of a temperature and humidity index (THI) [3,4].

The effect of heat stress was evaluated in a population of Brazilian Holstein cows using THI [3]. The authors used a total of 937,771 test day records from 3603 first lactations, and significant milk yield loss was observed with increasing THI. Recently, the genetic effects of heat stress on milk fatty acids in Brazilian Holstein cattle were studied [4]. The authors used 38,762 fatty acid records and 55,530 milk production records from 6344 Holstein cows within the first through sixth parity and showed that the fatty acid milk profile can change under heat stress conditions and that these changes may be genetically determined. On the other hand, there are also reports that there is no variation in heritability of milk traits along the THI in Holstein cows [5].

In addition, a proposal would be to evaluate animals in an environment that represents a situation of heat comfort and in an environment that represents a situation of heat stress based on the effect of SNP markers [6]. Thus, the aims of this study were: (i) to compare SNP markers associated with milk yield and quality traits as a function of two levels of a THI, one in the heat comfort zone and another one in the heat stress zone; (ii) to predict the direct genomic breeding values (DBV) based on the SNP effects estimated in each environment; and (iii) to verify if there is reranking of the animals accordingly with environment considered.

## 2. Material and Methods

### 2.1. Pseudophenotypes and THI from Previous Study

Information from milk yield (MY, in kg), somatic cell score (SCS), contents (%) of casein (CAS), and of saturated (SFA) and unsaturated (UFA) fatty acids in milk from Brazilian Holstein cows were used. In a previous study [5], the breeding values were predicted along a THI gradient using random regression models fitted using Legendre polynomials. In this previous study, 74,359 records of milk production and milk quality traits from 5220 cows were used.

Daily average THI for each herd was calculated in [5] using the following equation: THI = (1.8 × temp + 32) − [(0.55 − 0.0055 × rh) × (1.8 × temp − 26)], where temp is temperature in °C, and rh is relative humidity in percentage (daily averages). The THI values obtained were distributed in 167 different values, which ranged from 51.558 to 78.159 and presented, on average, the value of 68.052.

The breeding values obtained in the THI levels 59.426 (THI59) and 74.015 (THI74) were used as pseudo-phenotypes in the GWAS for the present study. Therefore, the breeding values of animals for each trait were considered pseudo-phenotypes in a bi-trait model where two levels of the THI were considered as “traits”.

The choice of these points is justified because: (i) there were a greater number of animals and measurements; (ii) they represent a point in heat comfort and a point in heat stress considering the threshold of 72 [7,8,9,10]; (iii) extreme values are excluded, which are not so robust in random regression analyses. Therefore, THI59 was considered as the heat comfort zone and THI74 as the heat stress zone.

In Appendix A, taken from the previous study [5], the number of records by THI value can be seen. Appendix A shows the distribution of THI values per year over the months used to determine the breeding values from the previous study. More details on the methodology used to obtain the breeding values and THI values can be found in [5]. 

### 2.2. Genotypes

Of the 5220 cows, 1157 were genotyped as follows: (i) 768 cows genotyped in a low-density panel (Illumina Bovine LD BeadChip 6k, Illumina, San Diego, CA, USA), which included 6909 SNPs; and (ii) 389 cows genotyped in a medium density panel (GeneSeek Genomic Profiler Bovine 50k, Neogen, Lansing, MI, USA), composed of 47,843 SNPs.

To enrich the genotypic data, they were submitted to imputation analysis. The reference population consisted of bulls, fathers of cows with genotypes, genotyped for 60,671 SNP on the platforms Illumina BovineSNP50 (Illumina, San Diego, CA, USA) or GeneSeek Genomic Profiler HD (Neogen Agrigenomics, Lexington, KY, USA). The imputation procedure was performed by the Animal Genomics and Improvement Laboratory (Agricultural Research Service, United States Department of Agriculture, USDA, Beltsville, MD, USA), using the findhap.f90 program [11]. This step was also carried out for the study of [12].

After imputation, there were 1149 cows with information on 60,671 SNPs. Quality control was performed, in which animals with call-rate less than 90%, as well as SNP markers located on the sex chromosomes with a call-rate less than 90%, with an allele frequency (MAF) less than 0.02, and with a deviation from Hardy–Weinberg equilibrium greater than 0.15 than expected were excluded. Finally, 1013 animals and 54,706 SNPs remained in the database.

Only the 1013 genotyped animals were considered in the subsequent analyses and their breeding values were selected from the previous study [5]. The descriptive statistic of the final data (i.e., breeding values used as pseudo-phenotypes) used in the present study is described in Table 1.

### 2.3. Model and GWAS

Genomic best unbiased linear prediction (GBLUP) was used under a bi-trait approach, with the traits considered as THI59 and THI74, and the breeding values considered the pseudo-phenotypes. The model considered was:(1)$$\left[\begin{array}{c} y_{1}\\ y_{2} \end{array}\right]=\left[\begin{array}{c} 1_{n}\mu_{1}\\ 1_{n}\mu_{2} \end{array}\right]+\left[\begin{array}{cc} Z_{1} & 0\\ 0 & Z_{2} \end{array}\right]\left[\begin{array}{c} \hat{a_{1}}\\ \hat{a_{2}} \end{array}\right]+\left[\begin{array}{c} e_{1}\\ e_{2} \end{array}\right]$$
where $y_{1}$ and $y_{2}$ are the vectors referring to the pseudo-phenotypes for THI59 (trait 1) and THI74 (trait2), respectively; $Z_{1}\;$ and $Z_{2}$ are the incidence matrices; $\hat{a_{1}}$ and $\hat{a_{2}}$ are the vectors to predict additive direct genomic effects of traits 1 and 2, respectively, with $\hat{a_{n}}\sim N\left(0,\;G_{0}⊗G\right)$, where $G_{0}$ is the additive direct genetic (co)variance matrix between the two traits, and **G** is the genomic relationship matrix, obtained based on the first method proposed by [13]; and $e_{1}$ and $e_{2}$ are the vectors of residual variance, with $e_{n}\sim N\left(0,\;R_{0}⊗I\right)$, where $R_{0}$ is the residual (co)variance matrix between the two traits and I is an identity matrix.

The (co)variance components were obtained by genomic restricted maximum likelihood method (GREML) using the AIREMLF90 program and the genomic breeding values (GEBV) were computed using the BLUPF90 program [14]. The SNPs effects were obtained from the GEBVs, using de postGSf90 program [14], using the equation described by [15]:(2)$$\hat{u}=DM^{\prime }{\left[MDM^{\prime }\right]}^{-1}\hat{a_{g}},$$
where $\hat{u}$ is the vector SNP effects; ***D*** is a weighting matrix for SNP, with ***D*** equal to identity matrix in the first iteration; ***M*** is the genotype matrix of the markers; and $\hat{a_{g}}\;$ is the vector with the predicted genomic breeding values.

The percentage of the explained additive variance of SNPs ($\hat{\sigma_{u}^{2}}$) was also estimated using the POSTGSF90 program [14]. The equation used for this step was [15]:(3)$$\hat{\sigma_{u,i}^{2}}=2p_{i}\left(1-p_{i}\right){\hat{u}}_{i}^{2},$$
where p is the allele frequency of SNP; and ${\hat{u}}^{2}$ is the SNP effect.

It is important to emphasize that the $\sigma_{u}^{2}$ was used solely for comparison between the two environments. Thus, it was not used to test the significance of the marker’s effect on the traits assessed. In addition, our goal was to compare the two environments and not to look for SNPs with significant effects. For this reason, no statistical tests were performed to identify significant SNPs.

The top 50 SNPs with the highest percentage of $\sigma_{u}^{2}$ were selected in THI59 and in THI74 for comparison. The SNPs that were not common to both environments were used to search for genes associated with the traits studied. The top 50 SNPs were selected for two reasons: First, because of the polygenic nature of the traits, no genes are expected to have a large effect. Second, because of the GBLUP assumption that the genetic model is an infinitesimal model, all SNPs have effects on the trait of interest. Thus, only the most extreme SNPs were compared.

The dispersion of the $\sigma_{u}^{2}$ in the quadrants defined by the truncation point (T) of the top 50 SNPs were also plotted for better visualization of the SNPs that were common and that were different between the two environments. The upper right quadrant shows the markers that have $\sigma_{u}^{2}>T$ for both environments. The lower left quadrant shows the markers with $\sigma_{u}^{2}<T$ for both environments. The lower right quadrant shows the markers with $\sigma_{u}^{2}>T$ for THI59 and $\sigma_{u}^{2}<T$ for THI74 (i.e., belonging to the top 50 for THI59 only). The upper left quadrant shows the markers with $\sigma_{u}^{2}>T$ for THI74 and $\sigma_{u}^{2}<T$ for THI59 (i.e., belonging to the top 50 only for THI74).

The DBVs were calculated for the animals in THI59 and THI74, using the PREDF90 program [14], based only on genotypes and SNP solutions obtained in the previous step, i.e.:(4)$$\left[\begin{array}{c} DBV_{1}\\ DBV_{2} \end{array}\right]=\left[\begin{array}{cc} M_{1} & 0\\ 0 & M_{2} \end{array}\right]\left[\begin{array}{c} \hat{a_{1}}\\ \hat{a_{2}} \end{array}\right]$$
where $DBV_{1}\;$and $DBV_{2}\;$are the direct genomic breeding values in THI59 (trait 1) and THI74 (trait 2), respectively; $M_{1}$ and $M_{2}$ are the genotype matrices; and $\hat{a_{1}}$ and $\hat{a_{2}}$ are the vectors of the estimated SNP effects.

Spearman rank correlation was performed to compare the classification of the top 10%, top 40%, and all animals between THI59 and THI74. 

## 3. Results

The Manhattan plots and the dispersion of the explained additive variance in the quadrants defined by the truncation point of the top 50 SNPs, by trait, can be viewed in Figure 1 and Figure 2. The truncation points T for MY, CAS, ECS, SFA, and UFA were 0.028, 0.030, 0.025, 0.031, and 0.030%, respectively.

There were no major differences in variation explained by SNPs between the two THI levels considered (Figure 1A and Figure 2A). The highest variance SNPs for the MY trait in THI59 are part of 21 chromosomes and represent 1.83% of the total explained additive variance. Chromosome 21 (BTA21) had the highest number of SNPs (8 SNPs), which represented 0.27% of the total additive variance. In THI74, the SNPs associated with MY are part of 21 chromosomes, totaling 1.83% of the explained additive variance and with a higher number also in the BTA21 (8 SNPs), which represented 0.27% of the total explained additive variance.

For CAS, the SNPs with the highest explained additive variance were distributed in 11 chromosomes in THI59 and in 15 chromosomes in THI74, which explained 1.91 and 1.85% of the variance, respectively. Still, for this trait, most of the SNPs are part of the BTA14, with 28 SNP and 0.99% of the variance for the THI59 and 24 SNP and 0.83% of the variance for the THI74, respectively.

Regarding the SCS trait, the BTA24 contained a greater number of SNPs with greater explained additive variance for both THI (7 SNP). These seven SNPs accounted for 0.20% of the variance in the THI59 and for 0.20% of the variance in the THI74. The top 50 SNPs associated with SCS were distributed in 20 chromosomes for both environments, with a total of 1.62 and 1.67% for THI59 and THI74, respectively.

For fatty acids, the top 50 SNPs with the highest explained variance for SFA are distributed in 15 chromosomes in THI59, representing 2.62% of the total explained additive variance, and in 17 chromosomes in THI74, representing 2.53% of the total additive variance explained. BTA14 contained more than 50% of the SNP with the highest variance in both environments (30 SNP), with 1.69% of the additive variance explained by the SNPs in THI59 and 1.63% in THI74. For UFA, the SNPs with the highest explained additive variance were located in 21 chromosomes for both environments, totaling 2.10% of the explained additive variance for THI59 and 2.09% for THI74. Of the 21 chromosomes, BTA14 had the highest number of SNPs (14 SNPs), which represented 0.65% of the explained additive variance for THI59 and 0.64% for THI74.

Overall, there was little difference between the top 50 SNPs with the highest explained additive variance for THI59 and THI74, with more than 80% of SNPs in common in both environments, for all traits (Figure 1B and Figure 2B). Additionally, the SNPs that were not common to both environments (n = 36) showed very close values of explained variance (%), as can be seen in Table 2. In Table 2, only 17 SNPs located near or within genes are shown.

Spearman’s correlations between DBVs of the top 10%, 40%, and all animals for the two environments evaluated were greater than 0.85 for all traits (Table 3), suggesting little or no change in the rank of the animals if they are evaluated at THI59 or THI74. The reranking plot of the top 10% of animals can be viewed in Figure 3.

Spearman correlations were high, and, even at the lowest correlations, e.g., CAS and SFA (0.88), the animals in the top 10% were almost the same, changing rank only, as can be seen in Figure 3. For MY, five of the 100 animals selected in the THI59 would not be the best if classified in the THI74. For CAS, 14 of the 100 animals selected in the THI59 would not be the best if classified in the THI74. For SCS, the classification of animals practically did not change, suggesting that, for this selection percentage, the choice of the best animals would not depend on the THI value. For SFA, 12 of the 100 animals selected in the THI59 would not be the best if classified in the THI74. For UFA, six animals of the 100 animals selected in the THI59 would not be the best if classified in the THI74. If the animals of the studied population are selected for MY, CAS, SFA, and UFA based only on the effect of their markers, at a selection ratio of 10% or 40%, the animals selected will be practically the same, whether selected in THI59 or in THI74.

## 4. Discussion

Five important dairy traits were included in this study, including the amount of saturated and unsaturated fatty acids in milk. These traits were evaluated using two environmental descriptors, one representing a heat comfort zone (THI59) and the other a heat stress zone (THI74). The effects of markers and direct breeding values were estimated in each of the environments.

Some SNPs associated with the studied traits, in both environments, are located within or close to genes known to act in the expression of traits evaluated. Although explained variance was not used as a criterion for selecting significant SNPs, the possible role of these genes in the expression of the evaluated traits was investigated.

An SNP associated with MY in this study are located within the gene *EXT2*, a candidate gene located in a region associated with fat production in Holstein cattle [16]. The gene *DNAJC3* has been associated with the occurrence of endoplasmic reticulum stress in the liver of dairy cows [17]. The LY6K gene has been associated with several milk production traits, such as 305-day milk, peak yield, and peak time in Holstein cattle [18].

Two SNPs with the highest explained additive variance associated with CAS for both the comfort and heat stress environments are located within the *FAM13A1* gene (family with sequence similarity 13, member A1), a gene that can act in the change of the protein content in bovine milk [19]. Considering the different SNPs for CAS (Table 2), the gene U6 snRNA has been associated with *sm-like protein LSM5*, that participate in RNA processing and to form part of the stress granule seen in stressed cells that contains mRNAs stalled in translation [20]. The gene *RBFOX1* was previously associated with subclinical ketosis in Jersey cattle [21], but no studies have been found that associate this gene with protein or casein content in milk from Holstein cows. Still, for CAS, among the 12 SNPs with the highest explained additive variance for both environments, three are within and one is close (195 kb downstream) to the *GHR* gene (growth hormone receptor). The *GHR* gene contains a mutation that has a strong effect on milk production and composition of cows, including yield and percentage of protein [22].

Few SNPs were different between environments for SCS and only one is located within a gene (*WDR59*). However, this gene has not yet been reported to be associated with somatic cell score in milk, nor with other traits related to milk production.

Several SNPs of greater explained additive variance associated with SFA and UFA are found within or close to the *DGAT1* (diacylglycerol O-acyltransferase 1) gene, as expected. The *DGAT1* gene is located on chromosome 14 of cattle and encodes acyl-Coa:diacylglycerol acyltransferase, an enzyme that catalyzes the last step of the synthesis of triacylglycerols [23], which are predominantly made up of fatty acids. The *DGAT1* gene is commonly associated with milk production and milk composition of cattle [24].

For SFA, one SNP is located within DGAT1, and 16 SNPs are located close to it (8 SNPs with 99 to 368kb upstream and 8 SNPs with 64 to 313kb downstream) and, for UFA, one SNP is located within DGAT1, and 8 SNPs are located close to it (99 to 332kb upstream). Specifically, for UFA, one SNP among the highest explained variance is located near the genes *SPP1* (osteopontin; 107kb downstream) and *ABCG2* (ATP binding cassette subfamily G member 2; 256kb downstream), and two other SNPs are also located close to the *ABCG2* gene (504 and 525kb upstream). In other studies, the *SPP1* gene has been expressed in mammary epithelial cells, has been found to be secreted in cow milk [25], and may be part of mammary gland development in mice [26]. In turn, *ABCG2* can act in the transport of cholesterol to milk [27].

In general, the differences between the percentages of additive variance explained by these SNPs and the differences in their classifications between environments were not large. Thus, even if the SNPs do not contribute equal proportions to the total additive variance in the two environments, the difference was very small.

Although with different methodologies, other studies also report genetic markers, different from those presented by the present study, which may be associated with a greater adaptation of dairy cows [28], and also associated with thermoregulation of these animals during exposure to heat stress [29]. In [28], GWAS was performed for the milk yield and a total of 39,000 SNPs. These authors used three groups of animals from Australia: a training population of approximately 62,000 Holstein cows and two validation groups, one consisting of 23,000 Holstein cows and the other consisting of 35,000 Jersey cows. The authors reported four SNPs associated with the change in milk production induced by heat stress. In [29], 625 Holstein cows and SNP previously associated with production, reproduction, and physiological traits involving heat stress were used. The authors reported the association of several SNPs with rectal temperature, respiratory rate, and sweating rate, as well as candidate genes associated with thermoregulation.

Although the amount of variance explained by SNPs was different between THI59 and THI74, these differences were very small. And although the markers were at very different rank positions in the different environments, the percentage of variance explained by them hardly changed.

Spearman’s correlations between DBVs of the top 10%, top 40% and considering all animals for the two environments evaluated were positive and high for all traits, indicating a strong association between the classification of animals between the two evaluated environments. If the selection of animals is based only on the effect of their SNP markers, the animals selected in THI59 will be the same in THI74, even selecting a small part of the animals, for example the top 10%.

As with the effects of the markers, no major differences in classification were found among the animals evaluated under the different environmental conditions studied. This little difference could be due to breeding values used as pseudo-phenotypes, which did not differ much between the two environments. In addition, in the previous study [5], no significant differences were observed in the pattern of the variance components and heritability coefficients along the environmental gradient. Several other factors may have contributed, for example, the THI levels used, the acclimation of the barns in which the cows were housed, and the polygenic nature of the traits themselves.

An important point is the definition of the THI values studied. In several studies, the THI threshold of 72 has been reported as the threshold between comfort and heat stress zone in dairy cows [7,8,9,10], and, in the present study, a THI value of 74 was used as the heat stress zone. Perhaps a value this close to 72 could have influenced the results, but, given the limited amount of data above the 72 limit, robustness of the analysis was prioritized.

The inclusion of a variable representing dairy cow resistance to heat stress in genetic evaluations is a promising proposal to reduce losses in milk production in the tropics and due to climate change. It is added that, although this study indicates some differences between different climatic environments, the variation between them was not very large. The performance of the animals used in the analyses could have been influenced by thermal management measures, such as a free-stall rearing system with the use of fans and sprinklers. Since the farms that participated in this study were striving to control and reduce heat stress, it is also possible that the hotter days were less stressful for the cows.

Although no major differences were found between the effects of SNP markers and direct breeding values obtained for animals in the two environments, this study is an important step toward understanding the genetic mechanisms involved in heat stress in the population studied. It is suggested that future studies be conducted with a larger number of data and animals at more extreme THI values, i.e., further from the threshold of the heat comfort and heat stress zones.

## 5. Conclusions

No major differences were found between the environments considered in terms of variation explained by the markers and reranking of animals for milk yield, casein percentage, somatic cell score, and percentage of saturated and unsaturated fatty acids in milk.

It is recommended that future studies be conducted using a larger number of animals and at more extreme THI levels.

## Figures and Tables

**Figure 1 genes-14-00017-f001:**
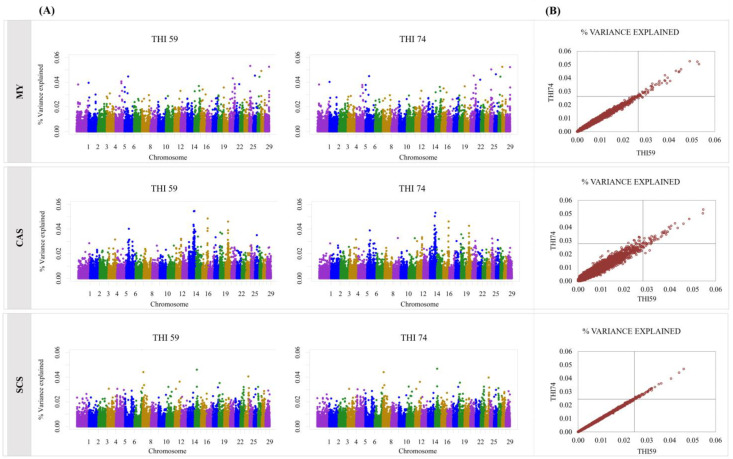
(**A**) Manhattan plots obtained by bi-trait analysis for THI59 and THI74 and (**B**) dispersion of the explained additive variance ($\sigma_{u}^{2}$) in the quadrants defined by the truncation point of the top 50 SNPs, for milk yield (MY), percentage of casein (CAS), and somatic cell score (SCS). In (**B**), for MY for example, the truncation point was 0.028%. Then the upper right quadrant shows the markers with $\sigma_{u}^{2}>0.028$ for both environments. The lower left quadrant shows the markers with $\sigma_{u}^{2}<0.028$ for both environments. The lower right quadrant shows the markers belonging to the top 50 for THI59 (i.e., $\sigma_{u}^{2}>0.028$ for THI59). The upper left quadrant shows the markers belonging to the top 50 only for THI74 (i.e., $\sigma_{u}^{2}>0.028$ for THI74).

**Figure 2 genes-14-00017-f002:**
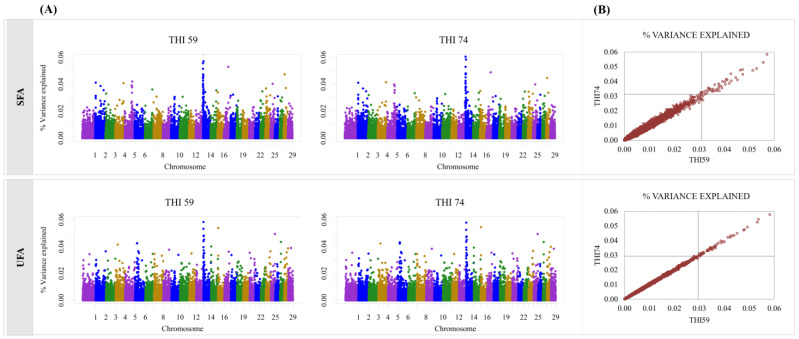
(**A**) Manhattan plots obtained by bi-trait analysis for THI59 and THI74 and (**B**) dispersion of the explained additive variance ($\sigma_{u}^{2}$) in the quadrants defined by the truncation point of the top 50 SNPs, for percentage of saturated fatty acids (SFA) and unsaturated fatty acids (UFA). In (**B**), for SFA for example, the truncation point was 0.031%. Then the upper right quadrant shows the markers with $\sigma_{u}^{2}>0.031$ for both environments. The lower left quadrant shows the markers with $\sigma_{u}^{2}<0.031$ for both environments. The lower right quadrant shows the markers belonging to the top 50 for THI59 (i.e., $\sigma_{u}^{2}>0.031$ for THI59). The upper left quadrant shows the markers belonging to the top 50 only for THI74 (i.e., $\sigma_{u}^{2}>0.031$ for THI74).

**Figure 3 genes-14-00017-f003:**
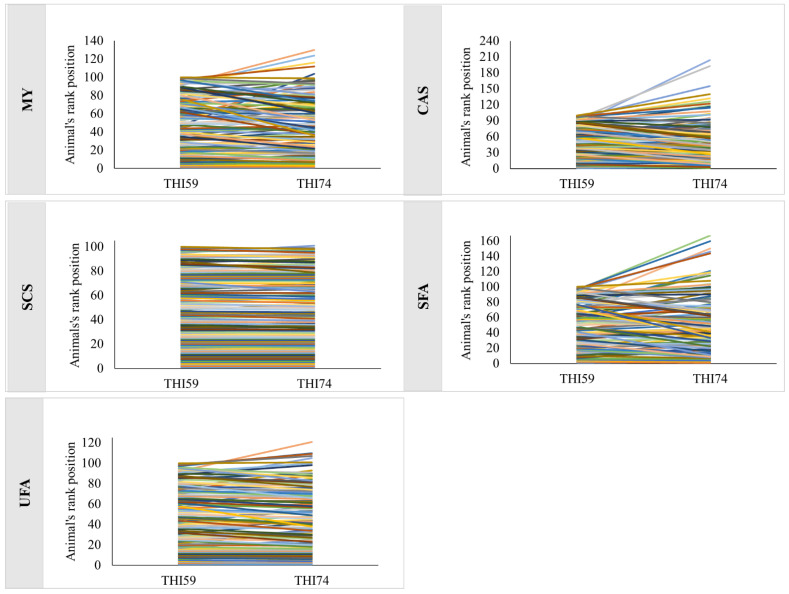
Reranking plot of the top 10% animals for milk yield (MY), percentage of casein (CAS), somatic cell score (SCS), and percentage of saturated fatty acids (SFA) and unsaturated fatty acids (UFA).

**Table 1 genes-14-00017-t001:** Number of observations (N), mean, standard deviation (SD), minimum (MIN) and maximum (MAX) values of breeding values used as pseudo-phenotypes for milk yield (MY), casein (CAS), somatic cell score (SCS), saturated (SFA) and unsaturated (UFA) fatty acids.

Traits		N	Mean	SD	MIN	MAX
MY (kg/day)	THI59	1013	1.13	2.13	−5.63	8.51
	THI74		1.00	2.01	−5.59	7.93
CAS (%)	THI59	1013	−0.04	0.11	−0.39	0.27
	THI74		−0.04	0.12	−0.41	0.28
SCS	THI59	1013	−0.13	0.53	−1.57	1.46
	THI74		−0.11	0.45	−1.33	1.22
SFA (%)	THI59	1013	−0.03	0.19	−0.67	0.57
	THI74		−0.02	0.17	−0.59	0.55
UFA (%)	THI59	1013	0.001	0.05	−0.17	0.13
	THI74		0.001	0.05	−0.19	0.15

**Table 2 genes-14-00017-t002:** Different SNPs in top 50 between THI59 and THI74, with the respective chromosome and position information (Chr:Pos), percentage of additive variance explained for THI59 (%$\sigma_{u}^{2}\;$THI59) and THI 74 (%$\sigma_{u}^{2}\;$THI74) environments, gene and gene position, for the traits milk yield (MY), casein content (CAS), somatic cell score (SCS), saturated (SFA) and unsaturated fatty acids (UFA).

SNP Name	Chr:Pos	$\%\sigma_{u}^{2}$ THI59	$\%\sigma_{u}^{2}$ THI74	Gene	Gene Position
MY					
BovineHD0800009461	8:31223530	0.03	0.04	*MPDZ*	31108126:31283422
BovineHD1200021758	12:77021076	0.03	0.04	*DNAJC3*	76958978:77024110
ARS-BFGL-NGS-103064	14:2754909	0.03	0.04	*LY6K*	2755205:2759123
ARS-BFGL-NGS-17362	15:75215172	0.03	0.02	*EXT2*	75074940:75265719
CAS					
BTA-147298	6:23183379	0.04	0.03	*SLC9B1*	23187624:23254634
BTA-75698	6:31570212	0.04	0.03	*PDLIM5*	31333660:31568270
ARS-BFGL-NGS-108245	6:96841919	0.02	0.03	*C4orf22*	96794834:97503371
BovineHD0900019961	9:71844127	0.04	0.03	*VNN1*	71832156:71851981
BovineHD0900019961	9:71844127	0.03	0.04	*U6 snRNA*	71844589:71844686
ARS-BFGL-NGS-85411	14:66339300	0.03	0.04	*SPAG1*	66319303:66406220
BovineHD1400018564	14:66400092	0.04	0.03	*SPAG1*	66319303:66406220
BTA-131179	16:75462069	0.04	0.03	*HSD11B1*	75463273:75533162
BovineHD2500000707	25:3293120	0.03	0.04	*ADCY9*	3233104:3344619
BovineHD2500001657	25:6561255	0.04	0.03	*RBFOX1*	4964122:6689485
SCS					
BovineHD1800000631	18:2324775	0.03	0.02	*WDR59*	2290822:2371069
SFA					
ss46527095	2:7321910	0.04	0.03	*COL3A1*	7317287:7356904
UFA					
BTC-033565	6:38286952	0.03	0.04	*MEPE*	38279473:38293661

**Table 3 genes-14-00017-t003:** Spearman’s correlations between direct breeding values of the top 10%, 40%, and all animals for the two environments (THI59 and THI74) for animals evaluated.

Traits *	Top 10%	Top 40%	All Animals
MY	0.90	0.95	0.99
CAS	0.88	0.93	0.98
ECS	1.00	1.00	1.00
SFA	0.88	0.94	0.98
UFA	0.97	0.99	1.00

* MY = milk yield; CAS = casein content; SCS = somatic cell score; SFA = saturated fatty acid; UFA = unsaturated fatty acid.

## Data Availability

The raw data cannot be made available because they are the property of the “Clínica do Leite” from “Luiz de Queiroz” College of Agriculture, University of São Paulo, and this information is commercially sensitive. The data supporting the results of this study are presented in the paper.

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
