# Peer review of "Comparison of Marker Effects and Breeding Values at Two Levels at THI for Milk Yield and Quality Traits in Brazilian Holstein Cows"

_genes, 2022, doi:10.3390/genes14010017_

Round 1
Reviewer 1 Report
Abstract: Add more information about the used statistical methods.
Materials and methods: Add more explanations about THI59 and THI74.
Results: Authors used tests to explain the statistical differences.
Table 4. the explanation about THI59*THI74 needs to be more correlated with studied traits.
Conclusion: Needs to be rewritten.
Reviewer 2 Report
The authors indicate "direct genomic genetic values", it should be "Direct genomic breeding values", or there is another kind of value? If it is correct, please explain.
In Materials and methods, authors include the procedure of imputation under the responsibility of an ARS researcher, he should be included in the contribution section and is part of the work
The major question, is: How do the authors define to compare the top (0.1%) SNP variance of THI59 and THI74? For the results and discussion, this threshold defines the rest of the document. I suggest getting the p values for both traits and comparing the significant SNP, as well as the genetic variance explained by the significant SNP.
Why do the authors estimate the Spearman correlation in the top 10% and not in all of the population? or probably an option could be to get deciles in the whole population for comparison.
Reviewer 3 Report
This a nice manuscript for analyzing the association between milk production traits and the SNP effects under two different THI levels. However, the manuscript lacks sufficient explanation of the results and comparative discussion with previous studies in discussion section, meanwhile, there didn’t have line numbers in the manuscript, all which makes the manuscript poorly readable.
Major Comments:
1).Both in abstract and conclusion, it said ‘the differences were very small for SNP in their explained variances between the two environments’, so what exactly those differences are, and which data or results leads you to this conclusion, and why.
2).Which part of your results showed the ‘marker effect and breeding values’ in your title. The manuscript must make enough explanation and clarification for those figures and tables in the results section, rather than letting the reader interpret and analyze the data themselves.
Minor comments:
1. Introduction
In the introduction section, it lacks a summary and analysis of previous studies in terms of THI, such as genetic variation underlying cow response to heat stress, genetic selection for thermotolerance/heat stress, etc.
2. Material and methods
2.1 Pseudophenotypes
‘In a previous study, the breeding values were predicted along a THI gradient using random regression models fitted using Legendre polynomials. In this previous study, 74,359 records of milk production and milk quality traits from 5,220 cows were used. The THI values obtained were distributed in 167 different values, which ranged from 51.558 to 78.159 and presented, on average, the value of 68.052. More details of the methodology can be found in [6].’
So, this paragraph depicted the previous data and related information. However, it didn’t clarify if the data used in this manuscript were from this previous study.
2.3. THI
How was THI calculated and what was its formula. Using data from previous studies, the references are required.
2.4. Model and GWAS
Page 4/13. For the DBV calculation, there were a1, a2 in the formula, but in the note, there were â1 and â2. Same erro in page 3.
3. Results
1).Figures and tables
For Figures, they don’t have explanation for the x and y axis. What do those charts/tables show and what they are trying to tell the reader, all those missing parts are exactly what the reader needs to know. Such as in Fig 3A, left graph can’t give much information. I don’t know what do they intend to tell us.
2).In Table2, If ‘Different SNPs’ were important, what were their genomic positional, or what were the genes or nearby genes of these different SNPs located to, and what functions of those genes, all which are needed in this table.
3). In Table3, there were genes in the table, but it didn’t show what the function of those genes was, which required here. Meanwhile, it only showed the information (%??) of THI59, not THI74.
4. Discussion
The discussion section did not explore or discuss the highlights or shortcomings of the results of this study, but rather mixed many descriptions of the study results, and the discussion section lacked the comparison between the results of this study and those of previous studies. For example, what the possible reasons were if this manuscript concluded that the SNP difference was small and the effect was low in the two environmental conditions (THI74, THI59), which were not adequately explained in the discussion section.
Round 2
Reviewer 2 Report
The authors made changes in the document that make it clear and easier for the reader to understand. Nevertheless, it is important to statistically justify the threshold of SNP to discuss. The "highly significant" could be 0.01% or 5%. The value is ambiguous.
The same case is when they select the top 10% of animals.
Reviewer 3 Report
1) In the revised manuscript, the Figure 1 wasn't the result of your current study, but result from previous one, so put figure 1 in the supplementary figure.
2) All Figures in the text needs enough explaination in the lengend.
3) In the results section. 'There were no major differences in variation explained by SNPs between the two THI................................. with more than 80% of SNPs in common in both environments, for all traits.' When you explaining your data, please add the figure number (Figure 2A or 3B) correspondingly, in order to let readers know what the expaination is for the figures. Because you never mentioned Figure panels in your text (Figure 3A, 4B, or 5A).
4) In previous reviewing, "In Table2, If ‘Different SNPs’ were important, what were their genomic positional, or what were the genes or nearby genes of these different SNPs located to, and what functions of those genes, all which are needed in this table." Please give your reasons If you didn't address the comments, like the function of the genes in coloumn forth wasn't provided.
